# Mental health of health care workers during and after the COVID-19 pandemic – A longitudinal cohort study

A. H. Ayesha Lavell[1,2], Jonne J. Sikkens[1,2], David T.P. Buis[1,2], Yvo M. Smulders[1,2], Christiaan H. Vinkers[3,4], Marije K. Bomers[1,2], Joeri K. Tijdink[5,6]*

1 Department of Internal Medicine, Amsterdam UMC location Vrije Universiteit Amsterdam, Amsterdam, The Netherlands, 2 Amsterdam Institute for Infection and Immunity, Amsterdam, The Netherlands, 3 Department of Psychiatry, Amsterdam UMC location Vrije Universiteit Amsterdam, Amsterdam, The Netherlands, 4 Amsterdam Public Health, Mental Health Program and Amsterdam Neuroscience, Mood, Anxiety, Psychosis, Sleep & Stress Program, Amsterdam, The Netherlands, 5 Amsterdam UMC location Vrije Universiteit Amsterdam, Department Ethics, Law and Humanities, Amsterdam, The Netherlands, 6 Department of psychology, UNPAD Universitas Padjadjaran, Bandung, Indonesia

* jk.tijdink@amsterdamumc.nl

## Abstract

Health care workers (HCWs) faced more stressors during the COVID-19 pandemic, potentially increasing depression, anxiety and post-traumatic stress disorder (PTSD). Insight into mental health dynamics and determinants in HCWs during the pandemic could help to maintain and improve mental health and resilience in future pandemics. In this longitudinal cohort study, HCWs received five surveys from November 2020 to March 2023 assessing self-reported symptoms of depression (PHQ-9), anxiety (GAD-7), PTSD (PCL-5), stress (PSS), burn-out (UBOS-EE), insomnia (ISI), resilience (RS) and work engagement (UWES). In addition to the longitudinal analysis, mental health symptoms were assessed in relation to possible predictors, e.g. patient care roles or prior SARS-CoV-2 infection. A total of 384 HCWs (95% of HCWs given consent) completed at least one survey, 326 (81%) completed two or more. Mental health significantly declined in December 2021 compared to November 2020, with mean increases of 1.16 (95% CI 0.73 to 1.58, d = 0.48), 0.79 (95% CI 0.41 to 1.17, d = 0.37) and 1.96 (95% CI 0.95 to 2.97, d = 0.35) on the PHQ-9 (range 0–27), GAD-7 (range 0–21) and PCL-5 (range 0–80), respectively, with similar results in multivariable analysis. Symptoms returned to November 2020 levels in March 2023. No differences were found regarding patient care roles, prior SARS-CoV-2 infection, years of work experience, or hospital workdays per week. Mental health significantly declined during the COVID-19 pandemic, after which mental health symptoms returned to baseline as the burden of COVID-19 patients decreased and public measures were lifted. This demonstrates this population's ability to successfully adapt to challenging experiences and emphasizes the need for support strategies tailored to the critical phases of any future healthcare crises.

**Data availability statement:** All relevant data are within the article and its supporting information files.

**Funding:** No funding was acquired for this project. The S3 cohort was supported by the Netherlands Organization for Health Research and Development ZonMw (S3 Study, Grant Agreement No. 10430022010023 to MKB) and the Corona Research Fund Amsterdam UMC (Reference No. 2020-013G to MKB).

**Competing interests:** The authors have declared that no competing interests exist.

## Introduction

Frontline health care workers (HCWs) face notably more stressors during pandemics, including increased mortality among their patients, moral dilemmas concerning the utilization of scarce resources, changing work environment, and the physical dangers of the increased risk of contracting an infection [1–3]. Previous studies have shown that this has led to a high prevalence of depression, anxiety, stress, post-traumatic stress disorder (PTSD) and insomnia [1,4–7]. Additionally HCWs working at intensive care units (ICUs) reported more burnout symptoms after the first wave of Coronavirus Disease 2019 (COVID-19) patients, compared with pre-pandemic times [8]. Identified risk factors associated with reduced psychological well-being include female gender, underlying health concerns, anxieties about family or personal infection, and shortages of personal protective equipment [9].

Less is known about how HCWs' mental health evolved during the COVID-19 pandemic and in its immediate aftermath. Previous studies also predominantly relied on binary outcomes derived from the arbitrary thresholds of mental health measurement instruments, potentially overlooking the nuances of continuous data [10]. Gaining insight into temporal trends of mental health during these times and its underlying determinants is expected to better support and care for the mental health of HCWs during future healthcare crises. This is of great importance given the fact that shortage of staff is already threatening many Western healthcare systems [2,11].

We therefore performed a longitudinal cohort study on the course of mental health symptoms in a cohort of HCWs during the height of the COVID-19 pandemic and its aftermath (2020–2023), focusing on trends over time and possible mental health determinants, such as differences between those working in direct patient care and those not working in patient care, or contracting a severe acute respiratory syndrome coronavirus 2 (SARS-CoV-2) infection prior to vaccination.

## Methods

### Ethics statement

The study was approved by both Medical Ethics Committees of the tertiary care centers of the Amsterdam University Medical Centers. Written informed consent was obtained from each participant.

### Study design and study population

This study is part of the larger Serologic Surveillance of SARS-CoV-2 cohort study among HCWs (S3 study). Selection and recruitment of study participants are described extensively elsewhere [12]. In short, a cohort of initially 801 hospital HCWs working in one of the two Amsterdam University Medical Centers, was set up in March 2020 and prospectively followed via both serologic measurements and surveys on COVID-19-related symptoms and exposure to COVID-19. During a follow-up visit in October 2020, study participants were asked to participate in the current mental health study. This comprised participation in five digital surveys, between November 2020 and March 2023, specifically focusing on mental health. Survey invitations

were dispatched to the email addresses provided by participants, and survey data were subsequently collected through Castor EDC [13].

## Pre-registration

The study was pre-registered at the Open Science Framework [14]. Before the start of the study, we hypothesized HCWs previously infected with SARS-CoV-2 or working in direct (COVID-19) patient care, would experience more mental health problems compared with those who were not infected or not working in direct patient care. As the pandemic progressed and vaccination efforts rapidly unfolded, with infections becoming increasingly prevalent and vaccines being widely distributed, our research focus evolved. We shifted towards conducting a longitudinal examination of the dynamics and determinants of mental well-being of HCWs over the course of the pandemic.

## Data collection and outcomes

**Primary outcomes.** We used the Dutch versions of the 9-item Patient Health Questionnaire (PHQ-9) [15], 7-item Generalized Anxiety Disorder (GAD-7) [16] scale and PTSD Checklist for DSM-5 (PCL-5) [17] to assess the severity of symptoms of depression, anxiety and PTSD, respectively. These tools are widely used and validated for measuring mental health symptoms and make it possible to compare our population with others.

The PHQ-9 and GAD-7 assess how often participants have experienced symptoms in the last two weeks, with responses scored on a 4-point Likert scale ranging from 'not at all' to 'nearly every day'. The PCL-5 consists of 20 items assessing the extent to which participants were bothered by problems in the past month on a 5-point Likert scale ranging from 'not at all' to 'extremely'. To maintain consistency with reference studies, we used the same Likert scales to ensure that the sum scores were comparable. For each of these instruments, sum scores were calculated by adding the individual item scores, with higher scores indicating greater severity of depression, anxiety, and PTSD symptoms, respectively.

To be able to compare our findings with other studies, sum scores were categorized based on cut-offs described in previous literature. PHQ-9 sum scores were categorized as follows: no/minimal (0–4), mild (5–9), moderate (10–14), moderately severe (15–19), or severe (20–27) depression [15]. GAD-7 sum scores: no (0–4), mild (5–9), moderate (10–14), or severe (15–21) anxiety disorder [16]. And PCL-5 sum scores: no PTSD below cut-off (0–33), or PTSD above cut-off (33–80) [18].

**Secondary outcomes.** We used the Perceived Stress Scale (PSS) [19], Utrecht Burnout Scale – subscale emotional exhaustion (UBOS-EE, the Dutch version of the Maslach Burnout Inventory) [20] and Insomnia Severity Index (ISI) [21] to assess the severity of symptoms of stress, burn-out/emotional exhaustion and sleep disturbance, respectively.

The PSS measures feelings and thoughts over the past month, with 10 items scored on a 5-point Likert scale ranging from 'never' to 'very often'. The UBOS-EE focuses on feelings related to work and consists of 8 items, scored on a 7-point Likert scale ranging from 'never' to 'always'. The ISI includes 7 items related to sleep experiences over the past two weeks, with responses scored on a 5-point Likert scale ranging from 'none' or 'very satisfied' to 'very severe' or 'very dissatisfied'. The sum scores of these instruments were calculated, with higher scores indicating greater severity of stress, emotional exhaustion, and sleep disturbance, respectively.

To assess resilience, we used the abbreviated 2-item version of the Connor-Davidson Resilience Scale (RS) [22], which asks participants to rate the extent to which two statements applied to them over the past month. Responses were given on a 5-point Likert scale, ranging from 'not at all true' to 'almost always true'.

Participants also received questions regarding work engagement (only in December 2021 and March 2023), assessed using the Utrecht Work Engagement Scale (UWES). The 9 item UWES (UWES-9) includes three subscales: vigor, dedication and absorption [23]. Items were scored on a 7-point Likert scale ranging from 'never' to 'daily'. Due to the very high correlations between the three subscales we used the total score, calculated by summing the scores of all items and

dividing by the number of total items [24]. Higher sum scores on the 2-item RS and UWES indicate better resilience and greater work engagement, respectively.

**Demographics and risk factors.** Questions about demographic characteristics (e.g. job function, marital status, living with children, years of experience in the field) were included in the first survey at the start of the mental health study. In October 2020, data was already collected within the larger S3 cohort study on participants' age, sex, professional roles and amount of exposure to COVID-19 patients, distinguishing between those specifically working in patient care with COVID-19 patients, those working in patient care with non-COVID-19 patients, and those not involved in patient care [12]. Whether participants were infected with SARS-CoV-2 prior to the widespread availability of vaccination for all HCWs in the Netherlands, was determined prior to the start of the mental health study and during the entire follow-up. It's important to note that this information was not available for all participants (Table 1).

## Statistical analysis

Continuous baseline variables were described as medians and interquartile ranges (IQRs) and categorical variables as absolute numbers and percentages. Survey outcomes, presented as sum scores per mental health item (assessed using the PHQ-9, GAD-7, PCL-5, PSS, UBOS-EE, ISI, RS and UWES), were treated as continuous outcomes and analyzed using univariable and multivariable linear mixed models, which accounts for both the within-subject dependence over time and occurrence of missing data due to subjects with missing survey responses. Time (survey time point) was included as a fixed effect to assess changes over time. A random intercept for each participant accounted for the aforementioned within-subject correlations. Multivariable models included time and possible confounders - age, sex, professional function (nurse/other patient care, doctor, or support personnel), marital status (with partner or single/other), and living with young children (<12 years old) - as fixed effects, based on prior literature [25,26]. Model residuals were checked to ensure normality assumptions. All subjects who responded to at least (part of) one survey, were included in the analysis.

Fixed effects estimates were extracted from the mixed model and divided by the residual standard deviation to calculate Cohen's d as a measure of effect size.

For subgroup analysis, to examine differences reflective of the specific characteristics of HCWs in patient care, we purposefully did not adjust for available HCW group variables. This approach aimed to identify potential underlying determinants of mental health tied to those distinct group characteristics.

Spearman's rank correlation coefficient was used to compare trends between mental health measures. Analysis were conducted in R version 4.2.1, using the lme4 package. Significance level was set at 0.05 (two-sided) or a 95% confidence interval that did not encompass zero. Graphs were created using GraphpadPrism version 9.5.1 and R version 4.2.1.

## Results

### Study population

A total of 459 participants attended the scheduled follow-up visit in October 2020, of whom 404 (88%) agreed to participate and gave written informed consent. Of those initially agreeing to participate, 384 (95%) responded to one or more surveys (supplementary S1 Table shows the response rates per survey), and 326 (81%) responded to at least two surveys. Overall, 76% of study participants were female and 68% worked in patient care (Table 1). Median age was lower in those who worked in patient care (37.6 vs. 50.6 years). The majority (62%) of participants employed in patient care were nurses while many of those not working in patient care were administrative personnel (34%). Nationwide vaccination campaigns started at the beginning of 2021 and the first healthcare workers were vaccinated between January and May 2021. By May 2021, 93 (24%) participants had been infected with SARS-CoV-2 prior to vaccination (Table 1).

**Table 1. Baseline characteristics.**

| | Overall (n = 384) | Working in patient care (n = 260) | Not working in patient care (n = 124) |
|---|---|---|---|
| **Age**, median (IQR) | 41.6 (31.6-54.6) | 37.6 (30.6-50.6) | 50.6 (39.1-57.6) |
| *Missing* | *4 (1.0%)* | *3 (1.2%)* | *1 (0.8%)* |
| **Sex**, female (%) | 292 (76.0%) | 197 (75.8%) | 95 (76.6%) |
| *Missing* | *2 (0.5%)* | *1 (0.4%)* | *1 (0.8%)* |
| **Marital status** | | | |
| With partner | 223 (58.1%) | 147 (56.5%) | 76 (61.3%) |
| Single | 106 (27.6%) | 70 (26.9%) | 36 (29.0%) |
| Other | 5 (1.3%) | 4 (1.5%) | 1 (0.8%) |
| *Missing* | *50 (13.0%)* | *39 (15.0%)* | *11 (8.9%)* |
| **Living with children** | 155 (40.4%) | 101 (38.8%) | 54 (43.5%) |
| *Missing* | *41 (10.7%)* | *31 (11.9%)* | *10 (8.1%)* |
| **Professional function** | | | |
| Nurse | 165 (43.0%) | 161 (61.9%) | 4 (3.2%) |
| Resident | 39 (10.2%) | 33 (12.7%) | 6 (4.8%) |
| Medical specialist | 53 (13.8%) | 51 (19.6%) | 2 (1.6%) |
| Other patient care | 14 (3.6%) | 14 (5.4%) | 0 (0%) |
| Administration/policy | 42 (10.9%) | 0 (0%) | 42 (33.9%) |
| Laboratory analyst/technician | 41 (10.7%) | 0 (0%) | 41 (33.1%) |
| Scientist | 13 (3.4%) | 1 (0.4%) | 12 (9.7%) |
| ICT, Pharmacy or other non-patient care | 17 (4.4%) | 0 (0%) | 17 (13.7%) |
| **Department** | | | |
| ED | 75 (19.5%) | 75 (28.8%) | 0 (0%) |
| ICU | 53 (13.8%) | 53 (20.4%) | 0 (0%) |
| Ward | 68 (17.7%) | 68 (26.2%) | 0 (0%) |
| Not in COVID-19 patient care | 188 (49.0%) | 64 (24.6%) | 124 (100%) |
| **SARS-CoV-2 infection prior to vaccination** | 93 (24.2%) | 77 (29.6%) | 16 (12.9%) |
| *Missing* | *112 (29.2%)* | *68 (26.2%)* | *44 (35.5%)* |

ED=emergency department, ICU=intensive care unit.

## Longitudinal mental health measurement in HCWs

**Primary outcomes.** At baseline (November 2020) the overall group reported low symptom scores of depression, anxiety and PTSD, with mean scores 4.38 (95% CI 3.98 to 4.78), 3.56 (95% CI 3.20 to 3.91) and 7.86 (95% CI 6.83 to 8.90) on PHQ-9, GAD-7 and PCL-5, respectively (Fig 1A-1C, S2 Table). Mental health symptoms significantly worsened in December 2021 compared to November 2020, with mean increases of 1.16 (95% CI 0.73 to 1.58, Cohen's d = 0.48), 0.79 (95% CI 0.41 to 1.17, d = 0.37) and 1.96 (95% CI 0.95 to 2.97, d = 0.35) on the PHQ-9, GAD-7 and PCL-5, respectively, with similar results in multivariable analysis (Fig 1A-1C and Fig 1J, S2 Table).

In March 2023, when the last COVID-19 measures (self-testing and isolation) in the Netherlands were lifted and many considered the pandemic to be over [27], mental health symptoms were comparable to November 2020 (mean difference of -0.54 [95% CI -1.01 to -0.07, d = -0.22], -0.50 [95% CI -0.92 to -0.08, d = -0.24] and -0.83 [95% CI -1.95 to 0.30, d = -0.15] on the PHQ-9, GAD-7 and PCL-5, respectively, yet not significant when adjusted for possible confounders; Fig 1A-1C and Fig 1J, S2 Table). Mean mental health survey scores at different time points, according to categories of confounding factors, are shown in S3 Table.

In November 2020, the proportions of HCWs scoring at or above the clinical cut-off values for depression, moderate to severe anxiety disorder or PTSD were 7.9% (27/342), 5.8% (20/342) and 1.8% (6/342), respectively (S4 Table). These proportions increased to 16.7% (38/228), 8.8% (20/228) and 5.7% (13/228) in December 2021, respectively, and subsequently decreased to 6.6% (11/166), 3.6% (6/166) and 3.6% (6/166) in March 2023 (S4 Table). Changes in depression, anxiety and PTSD symptom scores during follow-up, stratified by severity levels in November 2020, are shown in supplemental S2 Figs A-C.

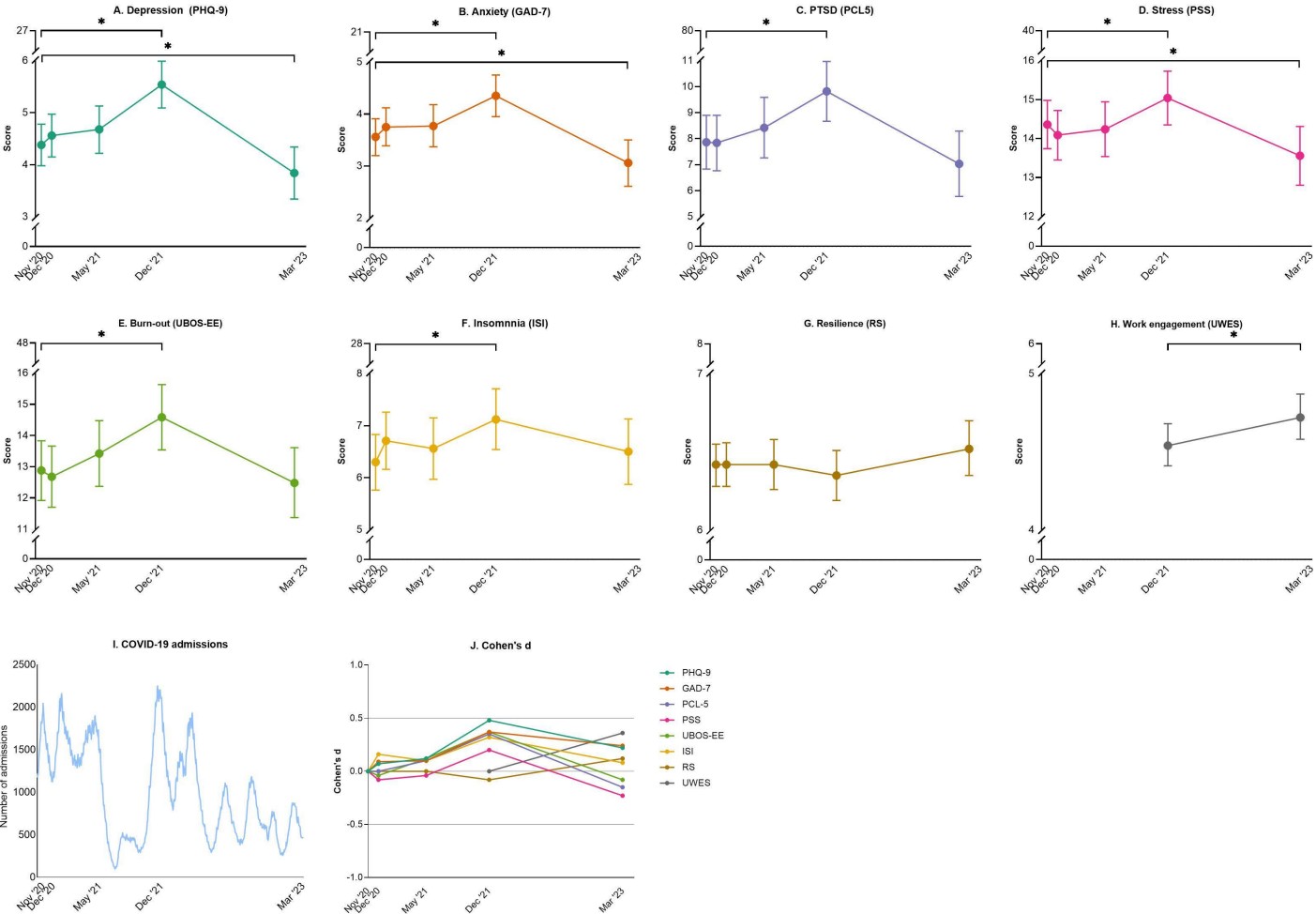

**Fig 1. Mean scores and 95% CI of mental health symptoms, COVID-19 admissions and effect size during follow-up.** On the x-axis the timeline is presented in days since the start of this study (November 2020). A-H: Mean scores of different mental health surveys and 95% confidence intervals (95% CIs) are presented by colored dots and vertical lines on different time points, obtained from the univariable linear mixed model. Lines with * indicate significant mean differences. For a more detailed visualization the y-axes in these figures are interrupted, complete scatterplots depicting individual sum scores of different mental health surveys combined with mean scores and 95% CIs are included in S1 Fig. I: Publicly available data from the national coordination center for patient distribution in The Netherlands ('Landelijk coördinatiecentrum patiënten spreiding') was used to depict the number of COVID-19 in-hospital admitted patients nationwide (continuous blue line) per day during study follow-up [28]. J: Colored dots indicate Cohen's d effect size of the mean difference in mental health survey outcomes compared to November 2020, calculated from the univariable linear mixed model assessing mental health in HCWs over time during the COVID-19 pandemic.

**Secondary outcomes.** Symptoms of stress, burn-out and insomnia also worsened between November 2020 and December 2021, and returned to baseline in March 2023 (Fig 1D-1F, S2 Table). Indicators of resilience did not significantly change during the follow-up period of our study (Fig 1G, S2 Table).

**Correlation between mental health outcomes.** The sum scores of the different surveys showed a positive correlation between the PHQ-9, GAD-7 and PCL-5 in November 2020 (rho between 0.73 and 0.76), and there was a positive correlation between the PHQ-9, GAD-7 or PCL-5 and the PSS, UBOS-EE or ISI (rho between 0.43 and 0.70, S3 Fig).

## Mental health in those working in patient care versus those not working in patient care

We hypothesized HCWs involved in direct patient care would experience more mental health symptoms compared with HCWs not working in patient care [14]. No substantial numerical nor significant differences in symptoms of depression, anxiety, and PTSD were observed between HCWs engaged in patient care and those not involved in patient care over the entire follow-up period (mean difference 0.04 [95% CI -0.59 to 0.67], 0.24 [95% CI -0.32 to 0.79] and 0.28 [95% CI -1.33 to 1.89] on the PHQ-9, GAD-7 and PCL-5, respectively, S5 Table).

Similarly, no differences were found in symptoms of stress, burn-out, insomnia, resilience, or levels of work engagement between HCWs working in patient care and those not working in patient care (S5 Table).

## Impact of COVID-19 work exposure, experience and SARS-CoV-2 infection status

We considered whether HCWs with increased exposure to COVID-19 patients experienced more mental health symptoms [14]. We found no significant differences in symptoms of depression, anxiety, or PTSD over the entire study period (S6 Table A) between HCWs working in designated COVID-19 patient care and those in non-COVID patient care. Similarly, no differences were observed based on SARS-CoV-2 infection status prior to vaccination, years of work experience, or number of days per week spent working in the hospital (S6 Table A).

In a subgroup analysis including only HCWs working in designated COVID-19 patient care, we found HCWs stationed on a COVID-19 nursing ward reported more symptoms of anxiety and PTSD than those working in the ED or ICU with COVID-19 patients, although the difference was not significant for the nursing ward - ICU comparison. The self-reported frequency of contact with suspected or proven COVID-19 patients was not associated with symptoms of depression, anxiety, or PTSD (S6 Table B).

## Work engagement

HCWs reported significantly higher work engagement in March 2023 compared to December 2021, as measured using the UWES questionnaire (mean difference 0.54 [95% CI 0.18 to 0.91], d = 0.36, Fig 1H and 1J, S2 Table). Specifically, in December 2021, HCWs involved in patient care more often reported doubts about continuing their career in healthcare compared to those not working in patient care (83/134 [61.9%] vs. 33/76 [43.4%], p = 0.01), see Fig 2 and S7 Table. This difference became smaller and non-significant in March 2023 (53/96 [55.2%] vs. 29/62 [46.8%], p = 0.38).

## Discussion

The aim of this longitudinal cohort study is to gain deeper insight into the course of mental health symptoms among HCWs during the peak of the COVID-19 pandemic and its aftermath, with a focus on trends over time and potential determinants. We found a significant increase in mental health symptoms (of depression, anxiety and PTSD) among hospital HCWs during the peak of the pandemic in the Netherlands (December 2021). Initially, in November 2020, mean scores were

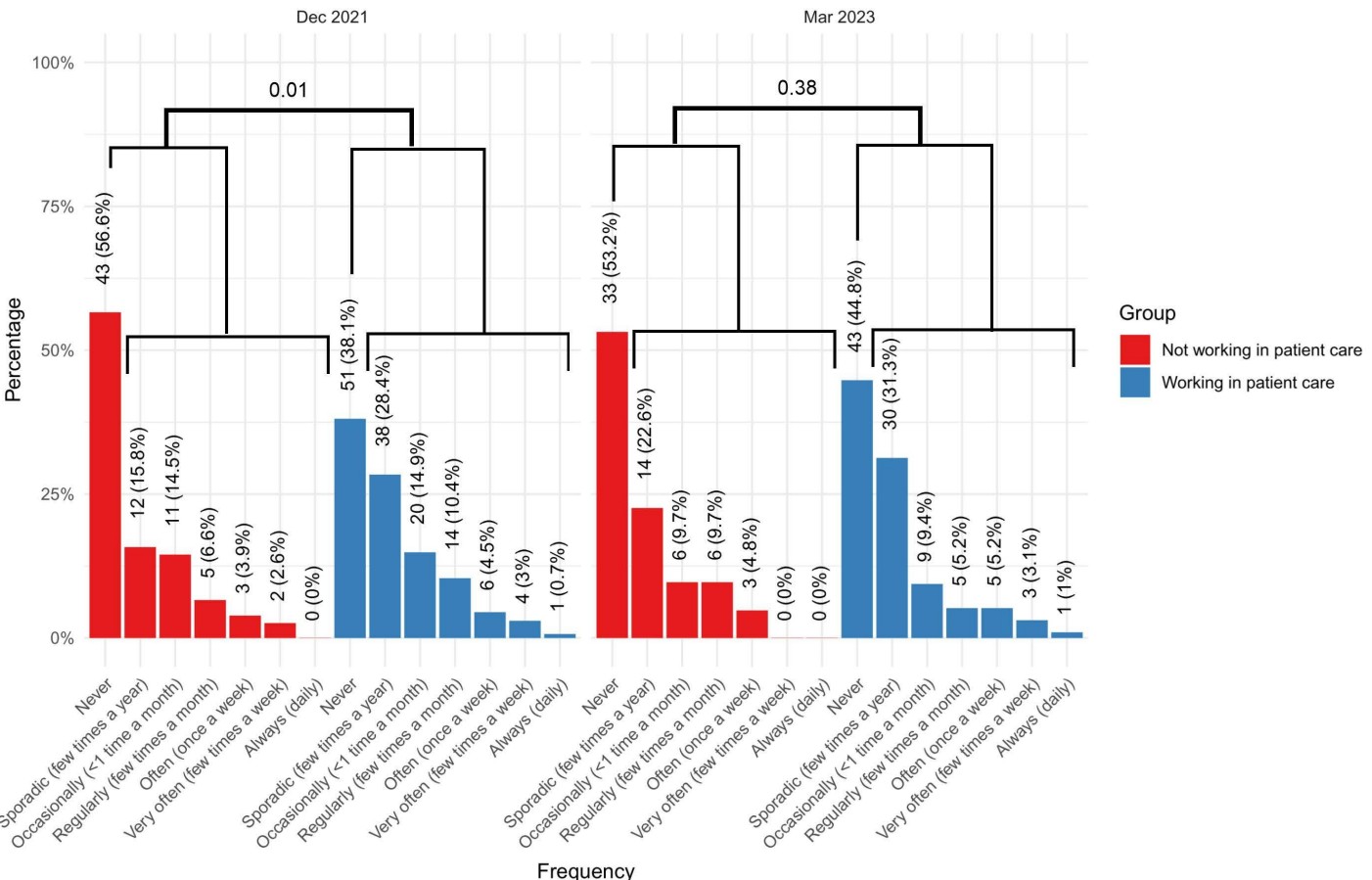

**Fig 2. Expressed doubts about continuing career in hospital health care.** Bar charts illustrating the number and percentage of participants reporting doubts about continuing their job in the hospital (from never to always) in December 2021 and March 2023, categorized by employment in patient care (blue) and non-patient care (red). Chi-square tests compare those reporting no doubts to those with any doubts (sporadic to always) within each group.

generally low, and by March 2023, scores had returned to these levels as most COVID-19 measures in hospitals and society had subsided.

Previous studies on HCWs' mental health during the COVID-19 pandemic have yielded conflicting results. A systematic review of 18 observational studies found that 12 reported a decline in mental health parameters, while 6 documented improvements [29]. This heterogeneity in findings may be attributed to differences in population, regional COVID-19 epidemiology and burden on healthcare system, timing, duration of follow-up, and survey types used to assess mental health. Many studies focused on frontline HCWs, whereas ours included HCWs not directly involved in patient care. Furthermore, numerous studies examined mental health solely during the initial pandemic wave or year [1,4,5,10]. Meanwhile, emerging evidence from 2021 and 2022 indicates a deterioration in mental health, although results vary across countries [25,30–35]. Consistent with our findings, another longitudinal study assessing symptoms of depression and anxiety on 2-item versions of PHQ and GAD, showed deterioration of mental health in German physicians during the course of the pandemic using continuous data up until 2022 [36].

The worsening of mental health symptoms in December 2021 could be attributed to the ongoing pandemic restrictions despite vaccine availability, which may have caused demoralization and a sense of hopelessness [37,38]. A resurgence in COVID-19 admissions, prolonged duration of exceptional healthcare conditions, and intensified public

polarization regarding the pandemic may have contributed as well [39,40]. Despite the worsening of symptoms during the pandemic, mean scores on symptoms of depression, anxiety and PTSD were notably low in our cohort. In November 2020, 7.9% and 5.9% of HCWs scored above the clinical cut-off values for depression and anxiety disorders, respectively. In comparison, the Netherlands Mental Health Survey and Incidence Studies (NEMESIS) reported a 9.8% prevalence for mood disorders and 15.2% for anxiety disorders in the general population between November 2019 and March 2022, with no differences in prevalence before and during the pandemic. Although it is important to note that differences between the general population and the HCWs described here may be influenced by the use of different diagnostic instruments and the potential for self-selection bias [41]. Some countries reported much higher rates, such as 41% depression, 50% anxiety, and 66% PTSD in Italy [25], highlighting variability between countries, cultures and health care systems [42,43]. Moreover, by March 2023, HCWs' mental health symptoms returned to November 2020 levels, indicating resilience. This resilience was also reflected in global data showing positive and negative emotions in 2023 returned to pre-pandemic levels [44].

Our study revealed no difference in mental health between HCWs in patient care and HCWs not in patient care, aligning with studies conducted in Belgium and the Netherlands [45,46]. However, a meta-analysis identified HCWs in direct COVID-19 patient care as having a higher risk of developing mental health problems compared to other HCWs [47]. Positive changes such as public recognition, moral duty, and a sense of autonomy might have contributed to the comparable mental health findings between these groups by providing psychological benefits that mitigated some of the stress and mental health issues typically associated with direct patient care [48,49]. No differences in mental health symptoms were observed based on SARS-CoV-2 infection status before vaccination, years of work experience, or the number of days working in the hospital. However, HCWs on COVID-19 nursing wards reported more anxiety and PTSD symptoms than those involved in COVID-19 patient care in the ED or ICU. These findings suggest that country-specific COVID-19 strategies, such as isolation measures, and increased tasks outside of the work environment (e.g. homeschooling children, caring for relatives), might have had a greater impact on mental health rather than factors related to working in health care.

Strengths of this study include the longitudinal follow-up of several mental health symptoms within the same group of HCWs. To our knowledge, ours is one of the few studies to assess mental health on a continuous scale in the same group of HCWs, including the aftermath of the pandemic. The correlations between specific questionnaires showed patterns consistent with theoretical similarities between the respective underlying constructs, supporting the validity of the results; e.g. we found stronger positive correlations between PHQ-9, GAD-7, and PCL-5, which measure related constructs expected to have similar but not identical outcomes. Our study also has limitations. Participants were required to be present in the hospital in October 2020 to provide written informed consent, potentially leading to selection bias. This may have led to an underestimating the overall mental health burden, as HCWs on sick leave after the first pandemic wave, could not be included [8]. Additionally, leave from the work environment during follow-up could have contributed to selective study discontinuation, potentially resulting in more missing outcomes in HCWs with more severe mental health problems. Additionally, we were not informed about pre-existing psychiatric conditions, which are linked to increased risks of depression, anxiety and PTSD among HCWs [50]. Lastly, our first assessment began a few months into the pandemic, so it cannot be considered a true baseline measurement of pre-pandemic mental health conditions.

## Conclusion

Mental health issues (symptoms of depression, anxiety and PTSD) among Dutch HCWs increased during the peak of the COVID-19 pandemic, but fully recovered to baseline levels thereafter. Our study found no differences in mental health based on exposure to COVID-19 patients, SARS-CoV-2 infection prior to vaccination, years of experience, or working hours, suggesting that non-work-related factors, such as country-specific measures and personal factors, had a more

substantial impact. Despite the generally low symptom scores and indications of resilience, the significant decline in mental health emphasizes the need for targeted support strategies during critical phases of public health crises. Future research should focus on understanding the drivers of HCWs' well-being and monitoring their mental health during crises to develop tailored interventions and prevent future declines.

## Supporting information

**S1 Table. HCWs participating in surveys.** The number of participants and percentage of health care workers (HCWs) who completed the surveys at different time points were calculated. In October 2020, participants in the larger S3 cohort were surveyed about their specific job title to determine whether they were working in patient care. Follow-up questions in December 2021 and March 2023 (as part of the mental health surveys) assessed if they had the same job or changed positions.
(XLSX)

**S2 Table. Results of longitudinal analysis of mental health in HCWs.** Crude model: Univariable linear mixed model assessing mental health outcomes in a cohort of 384 HCWs over time during the COVID-19 pandemic, with time as a fixed effect and a random intercept per individual. Adjusted model: Multivariable linear mixed model adjusted for age, sex, professional function (nurse/ other patient care, doctor, no patient care), marital status (partner, single/other) and living with children <12 years. Results are reported as means with 95% confidence intervals (CIs). Difference: difference in mean (and 95% CI) compared to baseline (November 2020). Cohen's d effect size for the mean difference. PTSD: Post-Traumatic Stress Disorder. Primary outcomes are reported in **bold** font, significant values in *italics*.
(XLSX)

**S3 Table. Mean scores and standard deviations per survey according to subcategories.**
(XLSX)

**S4 Table. HCWs with symptoms of depression, anxiety or PTSDs categorized by clinical cut-off scores.** Participants were categorized by the severity of symptoms of depression, anxiety and post-traumatic stress disorder (PTSD), based on clinical determined cut-off scores. Absolute numbers and percentages were subsequently calculated.
(XLSX)

**S5 Table. Assessing mental health symptoms in HCWs working in patient care versus those not working in patient care.** Univariable models assessing the association between symptoms of mental health of health care workers (HCWs) working in patient care and HCWs not working in patient care, over the entire study period. The difference is presented as mean with 95% confidence intervals (CIs). Primary outcomes are depicted in **bold** font.
(XLSX)

**S6 Table. Mean scores and 95% CIs according to SARS-CoV-2 infection status and work-related determinants.** # Adjusted for age, sex, marital status and living with children. *Adjusted for age, sex, professional function, marital status and living with children <12 years. During the study follow-up within the larger S3 cohort, we determined (by serologic surveillance and self-reported Nucleic Acid Amplification Tests [NAAT]-results of nasopharyngeal swabs), whether participants were infected with SARS-CoV-2 either prior to receiving vaccination or before the widespread availability of vaccination for all health care workers (HCWs) in the Netherlands (between January and June 2021). In October 2020, participants were surveyed about whether they worked directly in COVID-19 patient care, non-COVID-19 patient care or not working in patient care, as well as the number of years they worked in health care and the number of days per week they worked in the hospital. For those working directly in COVID-19 patient care we noted the amount of work-related

exposure to COVID-19 patients, i.e., by estimating the number of times HCW had been in the same room as a COVID-19 (suspected) patient in the previous four weeks (on a scale from none, to 1–5, 6–10, 11–25, 26–50, or more than 50 times), and their primary department of COVID-19 patient care (dedicated COVID-19 nursing ward, emergency department [ED], or intensive care unit [ICU]). Results are reported as mean with 95% confidence interval (CI).
(XLSX)

**S7 Table. Expressed doubts about continuing career in hospital health care.** In December 2021 and March 2023 participants received the question: "Are you considering to quit your job at the hospital?", with response options on a scale: never, sporadic (few times a year), occasionally (<1 time a month), regularly (few times a month), often (once a week), very often (few times a week), always (daily). Chi-square tests were used to compare individuals reporting never experiencing doubts with those reporting experiencing doubts (varying from sporadic to always), within both the groups working in patient care and not working in patient care.
(XLSX)

**S1 Data. De-identified participant survey data.**
(XLSX)

**S1 Fig. Individual survey sum scores during follow-up, combined with mean scores and 95% CIs.** S1 Figure A-I: On the x-axis the timeline is presented in days since the start of this study (November 2020). A-H: Means and 95% confidence intervals (CIs) of the survey sum scores are depicted per time point by small black horizontal lines and black vertical lines, obtained from the univariable model. A: On the y-axis sum scores per individual (teal dots) of symptoms of depression (assessed by PHQ-9) are presented, the dotted grey lines represent the different clinical cut-offs scores. B: On the y-axis sum scores per individual (orange dots) of symptoms of anxiety (assessed by GAD-7) are presented, the dotted grey lines represent the different clinical cut-offs scores. C: On the y-axis sum scores per individual (purple dots) of symptoms of post-traumatic stress disorder (assessed by the PCL-5) are presented, the dotted grey line represent the clinical cut-offs score. D: On the y-axis sum scores per individual (pink dots) of perceived stress (assessed by PSS), the dotted grey lines represent the different clinical cut-offs scores. E: On the y-axis sum scores per individual (green dots) of symptoms of burn-out (assessed by UBOS-EE) are presented. F: On the y-axis sum scores per individual (yellow dots) on symptoms of insomnia (assessed by ISI) are presented, the dotted grey lines represent the different clinical cut-offs scores. G: On the y-axis sum scores per individual (brown dots) of resilience (assessed by RS) are presented. H: On the y-axis sum scores per individual (grey dots) of work engagement (assessed by UWES) are presented, the dotted grey lines represent the different clinical cut-offs scores. I: Publicly available data from the national coordination center for patient distribution in The Netherlands ('Landelijk coördinatiecentrum patiënten spreiding') was used to depict the number of COVID-19 in-hospital admitted patients nationwide (continuous blue line) per day during study follow-up [28]. Abbreviations: min. = minimum sum score; mod. = moderate sum score, mod. sev. = moderately sever sum score; sev. = severe, max. = maximum; sub. = subthreshold; ave. = average.
(TIF)

**S2 Fig. Changes in depression, anxiety and PTSD symptoms according to baseline severity.** Participants were categorized by clinical determined cut-off scores indicating the severity of depression (PHQ-9), anxiety (GAD-7) and post-traumatic stress disorder (PTSD [PCL-5]), based on sum scores in November 2020. Individual data points, plotted with dots, indicate the distribution of PHQ-9, GAD-7 and PCL-5 sum scores at each time point. The mean scores for each severity category (based on November 2020) are represented by solid lines, with shaded areas around the lines depicting the standard error. The X-axis represents the different time points of data collection. The Y-axis indicate the sum scores, with higher scores reflecting greater severity of depression, anxiety and PTSD.
(TIF)

**S3 Fig. Correlation matrix (Spearman's rho).** Deltas of individual sum scores were calculated between follow-up time points for each participants. Spearman's rank correlation coefficient was used to assess the degree of correlation between the deltas of the different survey outcomes. Higher sum scores on the depression (PHQ-9), anxiety (GAD-7), post-traumatic stress disorder (PTSD [PCL-5]), stress (PSS), burn-out (UBOS-EE) and insomnia (ISI) surveys indicated decreased mental health. Higher sum scores on the resilience (RS) and work engagement (UWES) surveys indicate better resilience and work engagement.
(TIF)

## Acknowledgments

We would like to thank the HCWs of the Amsterdam UMC, who took the time to participate in our study during the pandemic. We would also like to thank Michiel Schinkel, for building the surveys in Castor EDC.

## Author contributions

**Conceptualization:** Jonne J Sikkens, David TP Buis, Christiaan H Vinkers, Marije K Bomers, Joeri K Tijdink.

**Data curation:** AH Ayesha Lavell, David TP Buis, Joeri K Tijdink.

**Formal analysis:** AH Ayesha Lavell, Jonne J Sikkens, Joeri K Tijdink.

**Funding acquisition:** Marije K Bomers.

**Investigation:** Jonne J Sikkens, Marije K Bomers, Joeri K Tijdink.

**Methodology:** AH Ayesha Lavell, Jonne J Sikkens, David TP Buis, Christiaan H Vinkers, Marije K Bomers, Joeri K Tijdink.

**Project administration:** AH Ayesha Lavell, David TP Buis.

**Supervision:** Jonne J Sikkens, Yvo M Smulders, Marije K Bomers, Joeri K Tijdink.

**Validation:** Jonne J Sikkens, Marije K Bomers, Joeri K Tijdink.

**Visualization:** AH Ayesha Lavell, Jonne J Sikkens, Marije K Bomers, Joeri K Tijdink.

**Writing – original draft:** AH Ayesha Lavell, Jonne J Sikkens, Marije K Bomers, Joeri K Tijdink.

**Writing – review & editing:** AH Ayesha Lavell, Jonne J Sikkens, David TP Buis, Yvo M Smulders, Christiaan H Vinkers, Marije K Bomers, Joeri K Tijdink.

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
