## [Decision Letter · Decision Letter 0]

2 Dec 2024

PMEN-D-24-00455

Mental health and resilience of health care workers during and after the COVID-19 pandemic – a longitudinal cohort study

PLOS Mental Health

Dear Dr. AH Ayesha Lavell,

Thank you for submitting your manuscript to PLOS Mental Health. After careful consideration, we feel that it has merit but does not fully meet PLOS Mental Health’s publication criteria as it currently stands. Therefore, we invite you to submit a revised version of the manuscript that addresses the points raised during the review process.

Reviewer 2 particularly draws your attention to the need to match the methodology to the objectives. This is important for the validity of the findings. 

We look forward to receiving your revised manuscript.

Kind regards,

Martin Mabunda Baluku, Ph.D.

Academic Editor

PLOS Mental Health

Journal Requirements:

Additional Editor Comments (if provided):

Reviewers' comments:

Reviewer's Responses to Questions

**Comments to the Author**

1. Does this manuscript meet PLOS Mental Health’s publication criteria?

Reviewer #1: Yes

Reviewer #2: Partly

2. Has the statistical analysis been performed appropriately and rigorously?

Reviewer #1: Yes

Reviewer #2: No

3. Have the authors made all data underlying the findings in their manuscript fully available (please refer to the Data Availability Statement at the start of the manuscript PDF file)?

Reviewer #1: Yes

Reviewer #2: Yes

4. Is the manuscript presented in an intelligible fashion and written in standard English?

Reviewer #1: Yes

Reviewer #2: Yes

Reviewer #1: I find this to be a very useful study which helps establish the effects of COVID-19 on the mental health of workers providing mental health services. The article is well written and the data is well presented.

Reviewer #2: I would like to thank the authors for the opportunity to read this paper. Mental health of HCWs is often neglected. I would like to commend the authors for trying to address this important issue in health, especially at the back of the COVID-19 pandemic where these important professionals were at risk of contracting COVID and carrying heavier workloads, among other workplace and societal ills. Insight into mental health dynamics and determinants in HCWs during the pandemic could help to maintain and improve mental health and resilience in future pandemics. Despite the merit, I feel that this manuscript could be improved before being considered by the journal. There seems to be a mismatch between the objective and the methodology - I think could the methodology be improved. For meaningful descriptive statistics, I suggest using regression sample. This will do away with varying sample sizes. I have attached my detailed comments.

**Do you want your identity to be public for this peer review?** For information about this choice, including consent withdrawal, please see our Privacy Policy

Reviewer #1: **Yes: ** Jorge A. Herrera Pino, M.D., Ph.D.

Reviewer #2: No

---

## [Decision Letter · Decision Letter 1]

25 Apr 2025

Mental health of health care workers during and after the COVID-19 pandemic – a longitudinal cohort study

PMEN-D-24-00455R1

Dear Dr. AH Ayesha Lavell,

We are pleased to inform you that your manuscript 'Mental health of health care workers during and after the COVID-19 pandemic – a longitudinal cohort study' has been provisionally accepted for publication in PLOS Mental Health.

Best regards,

Martin Mabunda Baluku, Ph.D.

Academic Editor

PLOS Mental Health

Reviewer Comments (if any, and for reference):

Reviewer's Responses to Questions

**Comments to the Author**

Reviewer #2: All comments have been addressed

Reviewer #3: All comments have been addressed

publication criteria?

Reviewer #2: Yes

Reviewer #3: Yes

3. Has the statistical analysis been performed appropriately and rigorously?

Reviewer #2: Yes

Reviewer #3: Yes

4. Have the authors made all data underlying the findings in their manuscript fully available (please refer to the Data Availability Statement at the start of the manuscript PDF file)?

Reviewer #2: Yes

Reviewer #3: Yes

5. Is the manuscript presented in an intelligible fashion and written in standard English?

Reviewer #2: Yes

Reviewer #3: Yes

Reviewer #2: I think you did a good job addressing my comments.

Reviewer #3: The manuscript demonstrates a high writing standard, flows coherently, and employs a rigorous methodology. The analyses conducted are suitable for a longitudinal study. The authors have cited both relevant and recent literature. I recommend considering the addition of a section on future research directions to facilitate the next steps in this important area of COVID-related research. Additionally, the authors should clarify if they performed any sensitivity analyses to illustrate how the treatment of missing data influenced their results. The study found no significant difference in mental health symptoms between patient-care and non-care workers. Discussing the effect size's implications, especially its clinical relevance, would be beneficial. In summary, this work makes a substantial contribution to the field.

**Do you want your identity to be public for this peer review?** For information about this choice, including consent withdrawal, please see our Privacy Policy

Reviewer #2: No

Reviewer #3: **Yes: ** Abigail Esinam Adade
